Phenotypical expression of reduced mobility during limb ontogeny in frogs: the knee-joint case

Ponssa Maria Laura 1 mlponssa@hotmail.com
Abdala Virginia 2
1 Unidad Ejecutora Lillo (UEL), CONICET-Fundación Miguel Lillo , San Miguel de Tucumán, Tucumán , Argentina
2 Cátedra de Biología General, Facultad de Ciencias Naturales e IML, UNT, Instituto de Biodiversidad Neotropical (IBN), UNT-CONICET , San Miguel de Tucumán, Tucumán , Argentina
Etchevers Heather
Electronic publication date: 2016 Feb 18
Publication date: 2016
Volume: 4
Electronic Location ID: e1730
Received 2015 Oct 15; Accepted 2016 Feb 2
Copyright: ©2016 Ponssa and Abdala
Copyright year: 2016
Copyright holder: Ponssa and Abdala
License: This is an open access article distributed under the terms of the Creative Commons Attribution License, which permits unrestricted use, distribution, reproduction and adaptation in any medium and for any purpose provided that it is properly attributed. For attribution, the original author(s), title, publication source (PeerJ) and either DOI or URL of the article must be cited.
License URL: https://creativecommons.org/licenses/by/4.0/

Keywords: Phenotypical expression, Reduced mobility, Knee-joint, Anurans, Development

Funding: CONICET PIP11220110100284 Funding was provided by CONICET (Grant number: PIP11220110100284). The funders had no role in study design, data collection and analysis, decision to publish, or preparation of the manuscript.

==============================
Movement is one of the most important epigenetic factors for normal development of the musculoskeletal system, particularly during genesis and joint development. Studies regarding alterations to embryonic mobility, performed on anurans, chickens and mammals, report important phenotypical similarities as a result of the reduction or absence of this stimulus. The precise stage of development at which the stimulus modification generates phenotypic modifications however, is yet to be determined. In this work we explore whether the developmental effects of abnormal mobility can appear at any time during development or whether they begin to express themselves in particular phases of tadpole ontogeny. We conducted five experiments that showed that morphological abnormalities are not visible until Stages 40–42. Morphology in earlier stages remains normal, probably due to the fact that the bones/muscles/tendons have not yet developed and therefore are not affected by immobilization. These results suggest the existence of a specific period of phenotypical expression in which normal limb movement is necessary for the correct development of the joint tissue framework.

Introduction

One of the most important environmental factors for the normal development of skeletal structures in tetrapods is movement, particularly during genesis and joint development (Abdala & Ponssa, 2012; Pitsillides, 2006; Nowlan, 2015). Most studies that deal with embryonic responses to changes in mobility have been performed on zebrafish, chickens, or mammals (Sullivan, 1966; Hall, 1975; Hosseini & Hogg, 1991; Müller, 2003; Pitsillides, 2006; Kahn et al., 2009; Shwartz et al., 2012; Nowlan et al., 2010a; Nowlan et al., 2010b; Nowlan, 2015). Conducting studies on vertebrates that undergo metamorphosis independent of maternal influences allows us focus on the effects of external environmental factors on mobility patterns and limb bud development. Abdala & Ponssa (2012) reported that the larvae of organisms that are free-living during development exhibit the same morphological responses to mobility reduction as embryos that undergo development in controlled environments, such as a uterus or a shelled egg, indicating a considerable degree of self-sufficiency during development. One explanation is that intrinsic genetic factors are responsible for initiating organogenesis, while extrinsic, epigenetic factors, including movement, exhibit a greater influence during later stages of development (Pitsillides, 2006). Another possible explanation is based on the concept of ‘critical periods’; specific periods of development during which, for example, the tissues, joints, etc., are sensitive to epigenetic factors (Hall, 1977; Pitsillides, 2006). Although the effects of immobilization on the development of vertebrates are relatively well known (Hosseini & Hogg, 1991; Pitsillides, 2006; Kim, Olson & Hall, 2009; Abdala & Ponssa, 2012; Nowlan & Sharpe, 2014a; Nowlan, 2015), the precise developmental stage at which the stimulus modification generates phenotypic modifications is still not well understood. In chicken embryos, it has been shown that fetal movements do not have an effect on joint shape until after cavitation has occurred (Nowlan & Sharpe, 2014a). In experiments analyzing mobility reduction in tadpoles, Abdala & Ponssa (2012) found that anomalies in tadpoles began at Stages 41–42, with no observable morphological consequences of mobility reduction during the initial stages of development. In this work, we focus on the developmental timing of phenotypical expression of mobility reduction during limb development in frogs. Our experiment was designed to evaluate (1) whether abnormal mobility affects the development of the limb tissues in the knee-joints and, if this is the case, we hypothesize that the effects of the stimulus would appear at the onset of ontogeny; or (2) whether mobility reduction affects only the assembly of the limb tissues in the knee-joints, in which case, we hypothesize that the phenotypical consequences would appear in Stages 40–42, as this is the ontogenetic phase in which tadpole limb tissues are mature and prepared to function (Manzano et al., 2012). Thus, we aim to understand the importance of movement on the development of the limb bud tissues during the formation of the knee-joint.

Material and Methods

One hundred and fifty Pleurodema borellii tadpoles were collected from temporary ponds in Lules and Yerba Buena (Tucumán, Argentina) (Field permit Res. 21/2012, Ministerio de Turismo) and sustained under laboratory conditions. We conducted five experiments with P. borellii tadpoles during the summer months (Dec–March) of 2011 and 2014. For each experiment we used three containers holding 1 L of water each; 9 g of agar were added to two of the containers (Abdala & Ponssa, 2012). Each repetition, therefore, consisted of one experimental, one experimental replicate and one control observation. In order to isolate the role of movement in knee-joint development, it was necessary to design an experiment that decreased movement of the tadpole’s limbs without producing other conditions adverse to growth and development.

The tadpoles were fed with locally available fish pellets ad libitum. It is important to clarify that while the experimental group of tadpoles experienced reduced mobility, they were able to reach the food pellets. Thus, both the control and experimental tadpoles fed normally and we were unable to detect differences in nutritional intake. The level of dissolved oxygen, measured with an oxygen meter (Hach sensION6), was 6.02 mg/L in the agar solution and 7.22 mg/L in the water control tanks. Successive measurements were made to insure that the difference in dissolved oxygen between the agar and the water was never more than 2 mg/L. In a typical fishpond, the critical oxygen concentration threshold is about 2 mg/L (Heargreaves & Tucker, 2002), therefore, by physiological measurements, the agar solution was a normally oxygenated medium. The agar solution had a density of 1.0 g/cm3, as water-colloids have the same density as water. Density of the medium was measured with a float-type densitometer. At 25 °C, water has a viscosity of 0.008 Pa/s. By adding agar, the medium viscosity was increased to 0.06 Pa/s, thereby imposing resistance on larval mobility. Viscosity was measured with an Ostwald viscometer. To avoid contamination, excessive solidification or a drastic decrease of dissolved oxygen in the agar medium, the colloid was replaced three times a week.

To gauge tadpole mobility, a 1-min digital video was recorded for 10 of the experimental and 10 of the control specimens selected at random (Abdala & Ponssa, 2012). The time the tadpoles spent moving was quantified and used as a measure of mobility. The videos were edited and analyzed using the program Version 6.0; Windows Movie Maker, Microsoft, Redmond, WA. Both control and experimental individuals were selected of different containers. A nested ANOVA was performed, considering the medium (agar or water) and container as nominal variables, and time of movement as measured variable. The data were converted to log10 to fit normality and homoscedasticity requirements and analyzed using an analysis of variance.

Ten Pleurodema borellii tadpoles at development Stage 34 were placed in each experimental and control container. In experiment A, 20 tadpoles (one experimental container, one experimental replicate container) were reared in agar medium; when they reached Stages 40–42, the agar solution was replaced by water and development continued until the juvenile stage. In experiment B, 20 tadpoles (one experimental container, one experimental replicate container) were reared in agar medium until the juvenile stage. In experiment C, 20 tadpoles (one experimental container, one experimental replicate container) were reared in water until Stages 40–42; the water was then replaced by the agar medium until the juvenile stage. In experiment D, 20 tadpoles (one experimental container, one experimental replicate container) were reared in water; when they reached Stage 40 the water was replaced by agar. Later, when the tadpoles reached Stage 42, the agar solution was again replaced by water and development continued until the juvenile stage. In experiment E, 20 tadpoles (one experimental container, one experimental replicate container) were reared in agar; when they reached Stage 40, the agar medium was replaced by water. When they reached Stage 42, the water was again replaced by agar and development continued until the juvenile stage. Additionally, 50 tadpoles were placed in plain water in containers similar to those holding the experimental tadpoles, and were monitored as controls to assess normal tadpole anatomy and development. Tadpole development encompasses both the growth and development of larval structures, and metamorphic changes. The Stage 40 is readily identifiable as the total length of the organism begins to diminish due to tail resorption and the larval mouth parts begin to break down (Gosner, 1960). The onset of metamorphosis, initiated at Stage 42 of development (Gosner, 1960), is externally evident as the forelimbs emerge, the mouth angle becomes anterior to the nostrils, labial denticles are lost and the horny beak disappears. The end of the metamorphic stages (Stage 46; Gosner, 1960) and the beginning of the juvenile period, defined as the stages between metamorphosis and sexual maturity, is recognizable by the development of the mouth and total tail resorption (Gosner, 1960).

To assess the mechanical effect of the agar medium on tissue differentiation, we conducted histological comparisons of the knee-joints of control tadpoles and experimental tadpoles that displayed clear phenotypic modifications. Formalin-fixed specimens were treated with 10% neutral buffered formalin and dehydrated with graded alcohol. Serial sections (6 µm thick) were cut on an MSE sledge microtome along the long axis of the limbs and at right and sagittal angles to the bone. Six sections were made every 1 mm and stained with Harris’s picrosirius haematoxylin (Totty, 2002). Histological sections of 21 experimental specimens and five control individuals were prepared. Histological samples of Pleurodema borellii obtained in experiments similar to the experiment B, described in Abdala & Ponssa (2012), were used to compare the tadpole anatomy. Histological data from the experimental tadpoles maintained in water through the initial developmental stages are not shown as they did not exhibit variation with respect to the control tadpoles.

Health status of the tadpoles was monitored throughout the experiment by examining the skin, oral disc, and limbs (Richards, 1962). In all cases, the tadpoles were observed to be healthy and no malformations associated with parasites or chemical compounds were detected (Meteyer, 2000). All the experiments were approved by the Bioethics Committee at the Facultad de Medicina, Universidad Nacional de Tucumán, Argentina (Res. No. 1206 2010).

Validity of experimental method

The conclusions drawn from this work depend upon the validity of our hypothesis: that mobility reduction is the sole cause of alterations in knee-joint development, and that there is no influence by unconsidered factors. The experiments were designed to avoid variability in the environmental conditions. All of the containers were located in the same room, with the same medium temperature, the same density, and the same dissolved oxygen. Nutritional intake and health, specifically related to parasites or chemical compounds, were monitored and appropriate for normal development throughout the experimental phases. The only aspect that varied was the viscosity of the living medium.

One of the consequences observed in the knee-joint tissues was a severe flattening of the hypertrophic chondrocytes. Quinn et al. (1998) stressed that mechanical loading through increased static compression is associated with a decreased cell radius in the direction of the compression, which would prove to be the case in the cartilaginous cells of the experimental tadpoles. The deformation illustrated by Quinn et al. (1998: Figs. 2C and D) resembles the present observations as well as results reported in a previous work (Abdala & Ponssa, 2012). This cell deformation suggests that the agar compresses the entire tadpole. Although compression is a collateral effect of living in the agar medium, damage observed in the cartilage is similar to that present in animals immobilized by drugs or denervation that did not suffer any mechanical load (Kim, Olson & Hall, 2009; Hosseini & Hogg, 1991). The areas of epiphyseal proliferation were the most affected regions of the long bone and the pathologies were similar to those previously described in the embryos of tetrapods, including mice, rats (Coutinho et al., 2002; Kahn et al., 2009) and chickens (Sullivan, 1966; Murray & Drachman, 1969; Hall, 1975; Hall & Herring, 1990; Quinn et al., 1998; Pitsillides, 2006), as well as in free-living tadpoles (Abdala & Ponssa, 2012). This consistent response is not surprising as the affected material was the same in all the cases: connective tissues (e.g., cartilages, bone, muscle, etc.).

Results

The amount of time spent in motion by the experimental tadpoles was no significantly different between containers within mediums (F(2,16) = 0.07; p > 0.5); but the movement was significantly lower in the experimental (agar medium) than that of control tadpoles (water medium) (F(1,16) = 15.92; p < 0.001). Of the 1 min recorded, the selected experimental tadpoles moved 5.63 ± 8.539 s (N = 10), while the controls moved for 33.2 ± 18.06 s (N = 10). The control individuals showed histological indications of normal tissue development (Figs. 1A, 1H, 1K, 1Q and 1B, 1G, 1L, 1R). Individuals from the most representative stages of each experiment were selected for detailed display.

Figure 1 External morphology and histological samples showing the knee-joint area, in successive developmental stages, of control and experimental (experiment A) specimens of Pleurodema borellii tadpoles.

(A–B) Control specimen, Stage 37–38: in the histological sample muscle tissue that is still not completely differentiated is observed; in the femur and tibiofibular epiphyses hyaline cartilage is evident, consisting of chondrocytes immersed in a basophile matrix composed of fibers and ground substance. Mesenchymal tissue surrounds the long bone epiphyses, forming regular tissue in the presumptive knee articulation area where the long-bone articular surfaces begin differentiation. Joint formation is evidenced by the interzone, a tight package of mesenchymal cells. The graciella sesamoid is embedded in dense connective tissue or tendon anlage (future tendinous tissue). (C–E) Experimental specimen, Stage 38: no evidence of malformation or abnormality is observed in the hind limb tissues. (F–G) Control specimen; (H–J) Experimental specimen, Stage 39: in the histological sample the cavitation process in the knee-joint led to a physical separation between the articular surfaces. In the cartilage of the long bones the resting, proliferating and hypertrophic zone can be distinguished. The osteochondral ligament and the articular lateral cartilage are differentiated in the epiphyses of the femur and the tibiafibula. The future tendinous tissue still appears like undifferentiated connective tissue. (K–L) Control specimen, Stage 42: the elements of the knee-joint, cavity and shape of the epiphyses, are already formed. (M–P) Experimental especimens, Stage 42 and Stage 43: the articular areas of the epiphyses are malformed, and the hyaline cartilage shows flattened cells with irregular borders. (Q–R) Control specimen, Stage 45: the histological sample was stained with Mallory trichome, allowing us to observe the mature tendon in blue. The tissues (cartilage, muscle and tendon) are completely mature at this stage. (S–T) Experimental specimen, Juvenile: note the severely damaged lateral articular cartilage in the detail, indicated by the black curved arrow. ct, connective tissue; e, epiphysis; F, femur; Fe, feet; fmgm, future gracilis major muscle; g, graciella sesamoid; HZ, hypertrophic zone; i, interzone; lac, lateral articular cartilage; ocl, ostechondral ligament; ta, tendon anlage; plm, plantaris longus muscle; PZ, proliferating zone; RZ, resting zone; ta, tendon anlage; TF, tibiafibula; T-F, tibiale and fibulare.

Figure 2 External morphology (A, C) and knee-joint histological section (B, D) of Pleurodema borellii tadpoles. Experiment B. External morphology and knee-joint histological section of tadpoles of Pleurodema borellii.

(A–B) Stage 36–37: Joint area without abnormality. (C–D) Stage 41: The articular cartilages are deformed; the chondrocytes are irregular and flattened. (E–F) Stage 44: Pathological menisci composed of hypertrophied and irregular cells. F, femur; lac, lateral articular cartilage; me, menisci; TF, tibiafibula.

Experiment A included specimens reared in agar medium from Stage 34 to 42. The agar solution was then replaced by water, and the tadpoles continued to develop until the juvenile stages. The external aspect of the specimens was highly modified by Stage 42 (Figs. 1M and 1N). Their joints exhibited abnormal angles and the descended hind-limbs were not supported by the pelvic girdle joint; the volume and tone of limb muscles displayed flaccidity (1M,N,S). Their histological structure before Stage 42 however, showed no evidence of malformation or abnormality (Figs. 1D, 1E, 1I and 1J). During Stages 42–43 on the other hand, the tadpoles began to exhibit clear signs of morphological pathology (Figs. 1O and 1P). The histological samples of the knee-joints of the experimental tadpoles showed articular cartilage with irregular boundaries and lateral articular cartilage that was thinner than in the control specimens (Figs. 1L, 1O and 1P). The chondrocytes that made up the hypertrophic zones of the long bones were highly malformed and presented extremely irregular borders (Figs. 1O and 1P); many of them were flattened (Figs. 1O and 1P). Upon completion of metamorphosis, the articular regions of the juvenile specimens exhibited irregular borders and the typical curved-shape of the epiphyseal articular area was not observed; the area of the lateral articular cartilage was also severely flattened (Fig. 1T).

Experiment B included tadpoles raised in agar medium from Stage 34 until the juvenile stage (Fig. 2). The phenotype showed modifications similar to those observed in experiment A. In experimental tadpoles at Stages 41–42, the articulation areas of the hind-limb long bones appeared deformed (Fig. 2C). The shape of the epiphyses was entirely modified. The cells of the osteochondral ligaments were not discernible and the articular cartilage was deformed. The ligaments and tendons of the knee-joint were not distinguishable, appearing as only a mass of pathological tissue between the femur and tibia-fibula epiphysis (Figs. 2D and 2F). The deformation was particularly remarkable in the menisci, which were completely composed of hypertrophied cells with irregular boundaries (Figs. 2D and 2F). The chondrocytes of the resting, proliferating and hypertrophic zones were highly irregular and flat compared to chondrocytes of normal specimens (Figs. 2D and 2F). For a more complete description of the effect of the agar medium on Pleurodema borellii tadpole development, see Abdala & Ponssa (2012).

Experiment C was composed of tadpoles reared in water between Stages 34–42 and in agar solution from Stage 42 onwards (to the juvenile stages). The tadpoles demonstrated normal development between Stages 34–42 (data not shown). After Stage 42 however, deformations became evident (Fig. 3). The tadpoles in Stage 44 had hind-limb long bones composed entirely of hypertrophied cartilage. This cartilage consisted almost completely of large lacunae that were highly irregular (Figs. 3B and 3C) compared to the cartilage of control specimens. The amount of interlacunar matrix was minimal and formed thin boundaries between the lacunae that resulted in a net-like appearance. The typical zones of differentiated chondrocytes in anuran long bones (reserve, hypertrophic and proliferation zones) were not distinguishable. No clear articulation area in the knee-joint was discernible (Figs. 3B and 3C). The cells of the osteochondral ligaments and the borders of the epiphyses were highly misshapen and all the articular cartilage was deformed with irregular borders.

Figure 3 External morphology (A) and knee-joint histological section (B, C) of Pleurodema borellii tadpoles. Experiment C.

(A–C) Stage 44: Note the hyaline cartilage composed of large poorly organized lacunae with irregular borders, and without an interlacunar matrix. The joint tissue is unrecognizable due to the high level of damage and there are no elemental differentiation in the articulation. F, femur; TF, tibiafibula.

Experiment D included tadpoles reared in water until Stage 40, in agar between Stages 40–42 and in water again from Stage 42 onwards (until juvenile stages). Tadpole development between Stages 34–40 was entirely normal (data not shown). After Stage 40 deformations became evident, congruent with the phenotypical modifications described above (Fig. 4).

Figure 4 Histological section of the femur epiphyses of Pleurodema borellii tadpoles. Experiment D.

(A–E) Stages 43–46: Note the damage in the articular area and in the lateral articular cartilage of the epiphysis of the femur. F, femur; lac, lateral articular cartilage; m, muscle.

Experiment E included tadpoles reared in agar until Stage 40, in water between Stages 40–42, and in agar again from Stage 42 onwards (to juvenile stages). Between Stages 34–40 the tadpoles displayed completely normal development (data not shown). After Stage 40 deformations again became evident, consistent with the phenotypical modifications described above (Fig. 5).

Figure 5 Histological section of femur epiphyses of Pleurodema borellii tadpoles. Experiment E

(A) Stage 40. (B) Stage 44. Note the completely irregular articular area and the injury in the lateral articular cartilage of the epiphysis of the femur. F, femur; lac, lateral articular cartilage; m, muscle; ocl, osteochondral ligament.

Pathological gross anatomical features in the joints

Figure 6 Experimental juvenile and metamorphic individuals showing gross anatomical consequences of mobility reduction, including inflected rigid fingers, fingers inflected in the palm, pronation of the elbow, rotation of the shoulder, abnormal extension of the knee.

We detected various body parts that were abnormally angled, bent or contractured. Some examples of the observed deformations were: extended and pronated elbows, digits bent towards the palm, flexed wrists, extended knees, and rotated shoulders. These defects severely modify the normal locomotory position of fore and hind limbs and therefore impact the ability to jump (Fig. 6 and Supplemental Information 1 and 2). In most cases we also detected a dermopathy throughout the body (Figs. 1N and 1S).

Discussion

Our experimental design permitted clear observation of the qualitative effects of immobilization on tadpole skeletal development. The results show that, although limb tissues of the knee-joint were exposed to the stimulus (reduced mobility) from the onset of ontogenetic development, they exhibit a normal phenotype until Stage 40–42 (Fig. 7), despite the fact that the elements of the knee-joint, cavity and shape of the epiphyses have already begun to form at Stage 38–39 (Manzano et al., 2012). The experiments allow us to postulate the existence of a specific phenotypical expression period between the Stages 40–42 (Gosner, 1960), in which abnormalities caused by immobilization begin to be observed in the knee-joint.

Figure 7 Scheme showing the results of the three experiments.

The red line indicates the appearance of abnormalities at Stage 40–42 in the five experiments.

Greater understanding of this specific period of phenotypical expression provides insight into the importance of stimuli during distinct points of developmental. For example, after this critical phase, the limb abnormalities provoked by reduced mobility are irreversible even if the tadpoles return to a normal environment. The most parsimonious explanation would be that the bones/muscles/tendons are not mature until Stages 40–42 and, therefore, mobility reduction has no effect on their prior development. In other words, once bones, muscles and tendons are fully formed at Stage 41 (Manzano et al., 2012), movement is essential for the maintenance of healthy tissues and the correct assembly and functionality of knee-joints in juveniles.

The results we obtained were unpredicted, as a correlation was expected between reduced mobility (stimulus) and tissue malformation (phenotypical expression). In Experiment A, for example, tissue modifications were anticipated to begin as early as Stage 34. From our data, we can extrapolate a new insight into the effect of reduced mobility on knee-joint formation: until Stage 40, limb tissue development seems to be isolated from mechanical stimuli engendered by movement. At this stage however, locomotion becomes critical to the assemblage, coordination and integration of the components of the knee-joint framework (Manzano et al., 2012; this work). One possible explanation is that limb tissue differentiation is strictly controlled by genetics and driven by the embryonic movement of the ischiadic nerve (Manzano et al., 2012; Nowlan, 2015), whereas normal tissue coordination and development is acquired through the epigenetic stimulus of movement. It is likely that embryonic movements are necessary to correctly shape bone ends through friction of the coupled surfaces (Murray & Selby, 1930; Hamburger & Waugh, 1940; Drachman & Sokoloff, 1966). Nowlan & Sharpe (2014a) and Nowlan (2015) showed that even though shape morphogenesis is advanced prior to cavitation of the hip joint, the absence of movement only affects hip joint shape morphogenesis after the development point at which joint cavitation has occurred. Both aspects, time of cavitation and phenotype modifications, coincide with our data: the joint shape is already formed before cavitation (Stage 39) and it is only affected by alterations in mobility in the posterior stages. Early morphogenesis, however, can be influenced by bending at the joint anterior to the cavitation joint, or by the stresses and strains of the differential growth of developing tissues (Henderson & Carter, 2002; Nowlan & Sharpe, 2014a). The phenotypical responses to our experimental design coincided with those observed in experiments using denervation and immobilization drugs (Hosseini & Hogg, 1991; Pitsillides, 2006; Kim, Olson & Hall, 2009); in both instances muscle contraction is impeded, preventing bending of the joints. This suggests that it is joint flexion, not the lack of muscle contraction, that promotes proper joint morphogenesis.

Muntz (1975) proposed four limb-development stages related to muscular maturity and locomotion: non-motile, pre-motile, motile, and fully functional. As our experiments begin at the end of the pre-motile stage, it can be assumed that in the earlier stages the ischiadic nerve stimulated limb bud mobility (Muntz, 1975; Manzano et al., 2012), and that this movement assisted normal knee-joint formation and hind-limb ossification. Our experimental design impeded the movement characteristic of the motile stage, and its absence dramatically affected further development of the limbs and their locomotory function. During the earliest stages of the motile phase (Stages 37–40), anurans showed an apparent insensitivity to external mechanical stimuli. We theorize that this passive phase is a consequence of the absence or immaturity of the elements involved in joint function: bones, muscles and tendons (Dunlap, 1967; Manzano et al., 2012). Thus, we propose that the severe complications caused by the reduction of movement in the early stages only begin to manifest themselves in Stages 40–42 (Gosner, 1960), as all the morphological alterations in the experimental tadpoles were visible only from Stage 40 onward. This is in accordance with the relatively late acquisition of the knee-joint tissue mechano-sensitivity, and the insensitivity to mechanical disruption during the early stages (Pitsillides, 2006). Stage 42 has been indicated as the beginning of the metamorphic stages (Gosner, 1960). During this critical phase, tadpoles lose their larval characteristics and begin to develop adult structures; the tail begins to resorb, larval feeding apparatus are replaced by the adult jaw and tongue and forelimbs and hind-limbs become functional. This period typically encompasses the passage from aquatic to terrestrial environments (McDiarmid & Altig, 1999). Thus, mobility reduction seriously affects the locomotor capacity of the froglet and its adaptation to the terrestrial habitat. Interestingly, the anterior damages produced by limited mobility were irreversible as tadpoles reared in agar medium until Stage 42 continued to display morphological alterations even when they were transferred to water after the froglet stage. Tadpoles reared in agar medium until metamorphosis and then transferred to water also exhibited anomalies in their locomotor abilities (Supplemental Information 1). Our data therefore allows us to propose the existence of a phenocritical phase that begins between the Stages 40–42. Limb joint mechano-sensitivity is acquired during this phase and alterations to its appropriate development will have lasting consequences. This demonstrates that finite periods during early development in frogs have an acute impact on the accurate arrangement of individual musculoskeletal components of the limb joints. Similar effects were described by Drachman & Coulombre (1962) who found that immobilization of fetal chicks through a brief period of the development provoked deformities, with the more advanced degrees of malformation associated with treatments initiated in later stages of development. The authors emphasize that even brief periods of immobilization can cause permanent deformities.

It is surprising that the response to this particular stimulus (reduced mobility) is independent of the ontogenetic environment of the individuals, suggesting that controlled fetal environments, such as a uterus or shell, have little influence in mitigating the effects of reduced mobility. Kahn et al. (2009) showed that movement-induced mechanical stimuli play a key role in the regulation of organ progenitor cells during joint development. They theorize that in some cases the lack of movement was offset by other genetic components that regulate joint development. Our data also suggest that normal limb movement is not a key factor in limb tissue development until the joint framework is assembled. After it is assembled, however, the lack of movement produced a phenotype typical of osteoarthritic joints. Alterations to metamorphic phenotypes arising from hostile larval environments can limit ecologically important performance capacities, including locomotion, and consequently influence food acquisition, predator avoidance, and dispersal capacities (Charbonnier & Vonesh, 2015). The dramatic consequences of mobility reduction on accurate joint development, and subsequently on locomotive capabilities, indicate this as an important topic for further study related to the survival possibilities of frogs.

Comparison of pathological features in our experimental frogs with abnormalities provoked by fetal akinesia

The anatomical modifications observed in the limbs of experimental frogs follow a pathological pattern coincident with the arthrogryposis condition, defined as congenital joint contractures in two or more areas of the body. The affected individual is unable to passively flex and extend the afflicted joints (Nowlan, 2015). The syndrome can be caused by neurogenic, myogenic, or connective tissue pathologies, or by environmental factors, such as a decrease in intrauterine movement (Kalampokas et al., 2012). The similarity of the pathological phenotypes produced by reduced mobility on the skeletal development of the frogs that we analyzed, chickens, and mice allow us to suggest that these are general features of tetrapods development. Taking this into consideration, frogs could also be recommended as a suitable animal system model for research regarding the effects of altered mechanical environments on the development of the vertebrate skeletal system (Nowlan, Chandaria & Sharpe, 2014b). Such research could shed new light on how to diagnose, prevent and treat different arthrogryposic syndromes.

Supplemental Information

Supplemental Information 1 Online resource I

Video showing juvenile specimen reared in agar medium until metamorphosis and then transferred to water. The individual display difficulty in its normal locomotion, is not able to stand on its limbs, and can only move erratically. The hindlimbs are placed laterally and have not a normal position for the jump.

Click here for additional data file.

Supplemental Information 2 Online resource II

Video showing juvenile specimen reared in water until metamorphosis, displaying normal locomotion.

Click here for additional data file.

Franco Pucci helped us in the production and interpretation of the histological data.

Additional Information and Declarations

Competing Interests

Author Contributions

Animal Ethics

Field Study Permissions

Data Availability

Virginia Abdala is an Academic Editor for PeerJ.

Maria Laura Ponssa conceived and designed the experiments, performed the experiments, analyzed the data, contributed reagents/materials/analysis tools, wrote the paper, prepared figures and/or tables, reviewed drafts of the paper.

Virginia Abdala conceived and designed the experiments, analyzed the data, contributed reagents/materials/analysis tools, wrote the paper, reviewed drafts of the paper.

The following information was supplied relating to ethical approvals (i.e., approving body and any reference numbers):

Comité de Ética de la Facultad de Medicina de la Universidad Nacional de Tucumán (UNT) Res. No. 1206 2010.

The following information was supplied relating to field study approvals (i.e., approving body and any reference numbers):

Ministerio de Turismo. Administración de Parque Nacionales. Res. 21/2012.

The following information was supplied regarding data availability:

The research in this article did not generate any raw data.

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
