# Peer review of "Phenotypical expression of reduced mobility during limb ontogeny in frogs: the knee-joint case"

_PeerJ, doi:10.7717/peerj.1730_

## Round 0.1 · original submission · Major Revisions

Both reviewers have offered detailed suggestions to improve this work, which they each appear to find interesting and worthwhile. We would therefore welcome your addressing all of their comments carefully, be it in rebuttal or in revision of the presentation or experimental design, and demonstrating these changes if you choose to re-submit.

In particular, it is important that other researchers could replicate your results, so it is only fitting that you try to replicate them yourselves, first, if possible, or explain why it is not. Please do specify the statistical assumptions as requested. You can reconcile the reviewers' injunctions to modify the discussion section by focusing on its most important arguments, such as the context in which your results contribute to the state of the art.

The manuscript should ideally be revised by a native English speaker for grammatical clarity and typographical errors before re-submission.

Reviewer 1 ·

Basic reporting

The study presents data showing that movement is necessary for the correct development of the knee joint in an anuran species from Argentina, and identifies a particular window during the larval phase critical for the correct development of this joint. Immobility during this sensitive period results in abnormal phenotypes.
The need for movement for the correct development of skeletal elements in vertebrates is a well-documented phenomenon, but the authors here contribute by identifying the stages at which movement is critical. It is implicit in the MS that this is a general feature of anuran development, but at this point this is an open question.
The MS writing style is in need of improvement, with numerous grammatical mistakes and a tendency for long sentences that are often confusing.
The discussion seems overly long, and perhaps could be reduced eliminating or greatly reducing the section comparing the results to fetal akinesia in other vertebrates. Likewise, the final section on validity of the experimental method seems rather out of place and perhaps would be best placed within the methods section. Instead, one would expect the end of the discussion to round up the paper and unambiguously remark the take-home message.

Experimental design

The methods need to better explain how the tadpoles fed (it is simply stated that pellets were ‘located achievable’), and whether there were likely differences between control and immobilized tadpoles in their feeding. Units need to conform to the International System (e.g. Li and gr are not appropriate abbreviations for liters and grams).
The description of the experimental procedures needs to be improved. There are inconsistencies in the reported number of tadpoles observed to estimate differences among treatments in tadpole movement (either 5 or 10 control tadpoles observed, lines 95 & 101). The statistical result reported in L101 should indicate the degrees of freedom, and it is unclear whether it corresponds to analysis of the frequency of movements or their duration, as they were both determined (L99). In any case, the authors should indicate if the variable analyzed met parametric assumptions, sine they seem to report ANOVA results.
The experimental design has a major flaw in that it is unreplicated. A single container per treatment was used, each container bearing 10 tadpoles. These 10 tadpoles in each treatment are thus pseudoreplicates and should be treated as such. At the very least, a random factor accounting for tank should be included in the linera models (i.e. mixed model glms). Unfortunately since there is a single tank per treatment, that may have the effect of undermining the statistical power to detect differences among treatments.
- If by ‘juvenile stage’ the authors refer to Gosner stage 46 (tail resorption), they should indicate it, to avoid ambiguity.
- How were the control tadpoles kept? Were they all together in a large tank? Were they distributed in similar containers to those holding the experimental tadpoles? (lines 114-115).
- On lines 130-131 the authors indicate that they compare the histology of the specimens from this experiment to previous specimens obtained in prior experiments, but it is unclear why this comparison is pertinent and if so, they need to explain how were those other tadpoles treated, not simply refer to a previous publication.
- The results should include a comparison among treatments of size and body mass at each developmental stages. It is well known that limb formation in anurans is highly plastic and that different growth trajectories alter limb shape and size. This information is important to determine the possible allometric changes occurring among treatments, which could affect knee development as it affects the degree of ossification. Actually, the statement on L208-209 regarding bone formation in stage 41 is not completely accurate since it does not take into account that bone formation is plastic and dependent on environmental conditions such as food availability or temperature. In other words, tadpoles at stage 41 could have very different degrees of ossification depending on the conditions experienced during larval development.

- The detailed histological analysis is remarkable, and is rather convincing of the abnormalities generated by immobilization at stages 41-42, although the issues regarding the experimental design persist.

- Figure 3. The last panel, labeled as ‘Juvenile’ is described in the figure legend as being stage 44, but it is clearly beyond stage 46, since it has no traces of tail remaining.
In general, the figures could be arranged somewhat differently. It would be useful to have both experimental and control tadpole images on the same figure, for ease of comparisons. Perhaps they could be arranged in stage-wise comparisons such as:

External morphology
Control ExpA ExpB ExpC ExpD ExpE

Stage 36-38
Stage 41-42
Stage 44-46

Histological view
Control ExpA ExpB ExpC ExpD ExpE

Stage 36-38
Stage 41-42
Stage 44-46

Likewise, the video shown provides no element for comparison. A second video of control juveniles moving would be needed to appreciate the differences in locomotion.

Validity of the findings

The results are rather qualitative and they are likely to be reliable, but the lack of replication is a major flaw. This could be partially solved statistically by including a random effect in the models, except that most of the results are visual inspections of the phenotype and there are no proper statistical analyses. The authors need to justify unambiguously to what extent each of the tadpoles analyzed in each treatment can be viewed as independent observations.

·

Basic reporting

.

Experimental design

.

Validity of the findings

.

Additional comments

See attached file

---

## Round 0.2 · Minor Revisions

Please respond briefly to the additional reviewer comments, particularly at the level of the statistical design and clarifying the utility of each uploaded video. Had you considered the possibility of adjusting the ANOVA analysis as suggested? The objection of the reviewer is phrased differently here, if that is helpful: http://www.biostathandbook.com/independence.html

I suggest modifying the first phrase of the Discussion in this way: "Our experimental design permitted clear observation of the qualitative effects of immobilization on tadpole skeletal development."

In a couple of minor typos:
- remove "-al" from "developmental" on page 9, line 189,
- page 10, line 214, introduce a hyphen at "tibia-fibula"
- page 14, remove the "s" from "denervations" line 293
- substitute "it is" on page 16, line 343 at "After its assembled".

Reviewer 1 ·

Basic reporting

In this revised version the authors have addressed most of my previous critiques and I have no further comments.

Experimental design

The design problem does not lie with the number of tadpoles used (which would be the main bioethical concern), but with how where they arranged. Most statistical analyses, and most definitely the ANOVAs used by the authors make the major assumption that the observations made (i.e. each data point) are independent from each other. This is clearly violated when several individuals from the same container are used as independent observations. This could be addressed by adding a random factor in the model grouping those observations coming from the same container, as I suggested in the previous review. Nevertheless, I realize that there is still a strong inertia within some disciplines of experimental biology (especially developmental biology) in ignoring the analytical consequences of these kind of designs.

The following clips seem identical:
peerj-6935-experiment_[review_response_.avi
peerj-6935-Online_Resorce_I_[experimental_froglet].avi

It is unclear what the reader learns from watching these two and the following video
(peerj-6935-Online_Resource_II_[juvenil].avi)

Validity of the findings

I stand by my previous statement. The number of individuals analyzed is reasonable, although the design suffers from pseudoreplication. Nevertheless, the results of this paper seem rather robust, they are largely qualitative and the treatment effects quite apparent.

Additional comments

No further comments

---

## Round 0.3 · accepted · Accept

We appreciate your attention to these last details. The English is not perfect in the new sentences you have added, but understandable.